# Tall Cell Variant versus Conventional Papillary Thyroid Carcinoma: A Retrospective Analysis in 351 Consecutive Patients

**DOI:** 10.3390/jcm10010070

**Published:** 2020-12-28

**Authors:** Alessandro Longheu, Gian Luigi Canu, Federico Cappellacci, Enrico Erdas, Fabio Medas, Pietro Giorgio Calò

**Affiliations:** Department of Surgical Sciences, University of Cagliari, S. S. 554, Bivio Sestu, 09042 Monserrato, Italy; gianlu_5@hotmail.it (G.L.C.); fedcapp94@gmail.com (F.C.); erdasenrico@libero.it (E.E.); fabiomedas@gmail.com (F.M.); pgcalo@unica.it (P.G.C.)

**Keywords:** tall cell variant of papillary thyroid carcinoma, conventional papillary thyroid carcinoma, thyroid surgery

## Abstract

Background: The aim of this retrospective study was to investigate clinical and pathological characteristics of the tall cell variant of papillary thyroid carcinoma compared to conventional variants. Methods: The clinical records of patients who underwent surgical treatment between 2009 and 2015 were analyzed. The patients were divided into two groups: those with a histopathological diagnosis of tall cell papillary carcinoma were included in Group A, and those with a diagnosis of conventional variants in Group B. Results: A total of 35 patients were included in Group A and 316 in Group B. All patients underwent total thyroidectomy. Central compartment and lateral cervical lymph node dissection were performed more frequently in Group A (42.8% vs. 18%, *p* = 0.001, and 17.1% vs. 6.9%, *p* = 0.04). Angiolymphatic invasion, parenchymal invasion, extrathyroidal extension, and lymph node metastases were more frequent in Group A, and the data reached statistical significance. Local recurrence was more frequent in Group A (17.1% vs. 6.3%, *p* = 0.02), with two patients (5.7%) in Group A showing visceral metastases, whereas no patient in Group B developed metastatic cancer (*p* = 0.009). Conclusions: Tall cell papillary carcinoma is the most frequent aggressive variant of papillary thyroid cancer. Tall cell histology represents an independent poor prognostic factor compared to conventional variants.

## 1. Introduction

Papillary thyroid carcinoma (PTC) is the most common malignant endocrine tumor. The prognosis for patients with PTC is almost the same as that of individuals who never had cancer, and only a few patients with PTC are affected by a biologically aggressive tumor [1,2]. The most common of PTC aggressive subtypes is the tall cell variant (TCV). TCV was first described by Hawk and Hazard in 1976 [3] and represents from 4% to 19% of all PTCs [4,5]. Several studies have demonstrated that this variant is generally underdiagnosed [6,7,8,9]. The definition of TCV includes the presence of a tumor whose cells are two to three times as tall as they are wide, eosinophilic cytoplasm, basilar-oriented nuclei, and the nuclear features of PTC [3,10,11]. TCV is generally considered a more aggressive variant of PTC and frequently has lymph node metastases and/or distant metastases, with a poorer prognosis [12,13,14] compared to conventional PTC (cPTC). The aim of this retrospective study was to investigate the clinical and pathological characteristics of TCV-PTC compared to conventional variants.

## 2. Materials and Methods

### 2.1. Study Design

This is a retrospective cohort study on patients who underwent thyroidectomy in our Unit of General and Endocrine Surgery (University of Cagliari, Cagliari, Italy) after the histopathological diagnosis of cPTC or TCV-PTC between January 2009 and December 2015. We identified cPTC with a classical variant, a follicular variant, and an oncocytic variant. Other biologically aggressive variants of PTC, papillary thyroid microcarcinoma, follicular carcinoma, medullary carcinoma, anaplastic carcinoma, secondary tumors, and tumors with a tall cell component < 50% were excluded. The patients were identified from a prospectively maintained institutional database, and those with incomplete or lost data at follow-up were excluded from the study. The patients were divided into 2 groups: those with a histopathological diagnosis of TCV-PTC were included in Group A, while those with a histopathological diagnosis of cPTC were included in Group B. Demographic data (sex and age), preoperative findings (cytological diagnosis and echographic features), surgical treatment (total thyroidectomy ± central compartment neck dissection ± modified lateral neck dissection), surgical outcomes (operative time and postoperative stay), histopathological findings, complications (hypoparathyroidism, recurrent laryngeal nerve injury, postsurgical cervical hematoma, wound infection, and chylous fistula), and follow-up data (local recurrence and distant metastases) were recorded.

### 2.2. Preoperative Evaluation

For each patient, preoperative assessment consisted of free triiodothyronine (FT3), free thyroxine (FT4), and thyroid-stimulating hormone (TSH) blood measurements; high-resolution ultrasound (US) of the neck; and fibrolaryngoscopy for assessment of vocal fold mobility. In the case of suspicious nodules, US-guided fine-needle aspiration cytology was performed.

### 2.3. Surgical Procedure

All operations were performed under general anesthesia by the three most skilled endocrine surgeons of our unit. Recurrent laryngeal nerves and parathyroid glands were systematically searched and identified. Intraoperative nerve monitoring (IONM) was routinely used to facilitate nerve identification and to confirm its functional integrity. Hemostasis was mainly achieved using energy-based devices. One or two closed-suction drains were placed below the strap muscles. The cervical linea alba and platysma were sutured with absorbable sutures, and the skin was closed by a continuous intradermal suture. The duration of the surgical procedure, from skin incision to skin closure, was estimated in minutes.

### 2.4. Postoperative Management and Follow-Up

The serum calcium and PTH levels were assayed pre- and postoperatively. Postsurgical hypoparathyroidism was defined as PTH < 10 pg/mL following the operation (normal range = 10–65 pg/mL). Permanent hypoparathyroidism was defined as PTH concentrations below the normal range for more than 12 months. In case of suspected recurrent laryngeal nerve injury, a fibrolaryngoscopy was performed to assess vocal cord mobility. Postoperative radioactive iodine therapy (RAI) was administered, according to the 2009 American Thyroid Association guidelines, in case of gross extrathyroidal extension, primary tumor size greater than 4 cm, distant metastases, or selected patients with a primary tumor, ranging from 1 to 4 cm, confined to the thyroid gland but with a significant risk of recurrence. Follow-up consisted of neck US examination and dosage of thyroglobulin (Tg) and thyroglobulin antibodies (TgAb) levels every six months during suppressive L-thyroxine treatment (a serum Tg level of 0.2 ng/mL was considered as undetectable). In patients with suspicious recurrence, a whole-body ^131^I scanning after recombinant human thyrotropin (rhTSH) was performed. The diagnosis of disease recurrence in the cervical lymph nodes was based on serum Tg level monitoring, US-guided fine-needle aspiration cytology (FNAC), and Tg washing of FNAC aspirates.

### 2.5. Statistical Analysis

Statistical analyses were performed with MedCalc^®^ (Ostend, Belgium) 19.1.3. The Fisher exact test or chi-squared test was used for categorical variables, and the t-test for continuous variables. The Kaplan–Meier method was used to analyze disease-free survival curves. *p*-values < 0.05 were considered statistically significant.

## 3. Results

As reported in Table 1, 351 patients were included in this study: 35 (9.97%) in Group A and 316 (90.03%) in Group B. Women were more numerous than men in both groups. The mean age was 49.3 ± 18.2 years in Group A and 50.6 ± 14.9 in Group B (*p =* 0.6). An indeterminate or suspicious nodule was identified in 8 (22.8%) cases in Group A and 131 (41.4%) in Group B (*p =* 0.04), whereas a carcinoma was diagnosed in 14 (40%) patients in Group A and 24 (7.5%) in Group B (*p* < 0.0001). Suspicious echographic features were found in 14 (40%) patients in Group A and 119 (37.6%) in Group B (*p =* 0.8). Benign thyroid disease (multinodular goiter and hyperthyroidism) was associated with thyroid cancer in 10 (28.5%) patients in Group A and 87 (27.5%) in Group B (*p =* 0.8).

All patients underwent total thyroidectomy; central neck compartment lymphadenectomy was associated with thyroidectomy in 15 (42.8%) patients in Group A and 57 (18.03%) in Group B (*p* < 0.001), whereas modified lateral neck dissection was performed in 6 (17.1%) patients in Group A and 22 (6.9%) in Group B (*p =* 0.04). The mean surgical time was 121 ± 29.01 min in Group A and 100 ± 28.91 min in Group B (*p =* 0.0004). The mean postoperative stay was 2.74 ± 0.97 days in Group A and 2.68 ± 0.93 in Group B (*p =* 0.7) (Table 2).

The mean tumor size was 2.69 ± 1.27 cm in Group A and 2.22 ± 1.25 cm in Group B (*p =* 0.03); thyroid weight was 31.9 ± 42.99 g in Group A and 33.91 ± 42.37 g in Group B (*p =* 0.7). Multicentric cancer was found in 11 (31.42%) patients in Group A and in 110 (31.64%) in Group B (*p =* 1), angiolymphatic invasion in 6 (17.14%) patients in Group A and in 16 (5.06%) in Group B (*p = 0*.01), parenchymal invasion in 13 (37.14%) patients in Group A and in 56 (17.72%) in Group B (*p = 0*.004), and extrathyroidal extension in 11 (31.42%) patients in Group A and in 16 (5.06%) in Group B (*p* < 0.0001). Cervical lymph node metastases were found in 16 (45.71%) patients in Group A and in 42 (13.29%) in Group B (*p* < 0.0001) (Table 3).

In Group A, postoperative hematoma occurred in one (2.85%) patient, transient recurrent laryngeal nerve palsy in one (2.85%), transient hypoparathyroidism in 11 (31.42%), and permanent hypoparathyroidism in three (8.5%), whereas wound infection, permanent recurrent laryngeal nerve palsy, and chylous fistula did not occur. In Group B, postoperative hematoma occurred in four (1.26%) patients, transient recurrent laryngeal nerve palsy in four (1.26%), transient hypoparathyroidism in 87 (27.53%), and permanent hypoparathyroidism in 20 (6.3%), whereas wound infection, permanent recurrent laryngeal nerve palsy, and chylous fistula did not occur (Table 4).

The mean follow-up was 79.4 ± 25.9 months in Group A and 98.4 ± 26.5 in Group B (*p =* 0.09). Local recurrence affected 6 (17.1%) patients in Group A and 20 (6.3%) in Group B (*p =* 0.02); 2 (5.71%) patients in Group A developed distant metastases, whereas distant metastases did not occur in Group B (*p =* 0.009) (Table 5).

Five-year disease-free survival was 82.3% in Group A and 92.8% in Group B (*p =* 0.0018) (Figure 1).

## 4. Discussion

Only a few patients with PTC are affected by a clinically aggressive tumor, and the most common of these subtypes is TCV-PTC, which was first described by Hawk and Hazard in 1976 [3]. The definition accepted widely by pathologists includes the presence of a papillary tumor whose cells are at least twice as long as they are wide [3,11]. Currently, this variant is underdiagnosed: several studies have demonstrated that when cases diagnosed as cPTC were reviewed by endocrine pathologists, 1–13% of the tumors were identified as TCV-PTC [6,15,16]. One of the obstacles to the correct diagnosis of TCV-PTC is the lack of consensus as to how much of the tumor must be composed of tall cells to make a diagnosis. The cutoff varies from one institution to another, from 30% to 70% [10,17,18], and any tumor that contains a smaller percentage of tall cells than the institutional cutoff is classified as PTC with tall cell features [18]. In our institute, tumors with less than 50% tall cells are excluded by the diagnosis of TCV-PTC, and they were excluded from this study because the implications of the presence of a small number of tall cells in a thyroid tumor are currently debated [6,7,8,9]. However, some authors found an association between PTC with tall cell features and a poorer prognosis than cPTC [19,20,21,22]. FNAC is the most useful tool in the preoperative diagnosis of papillary carcinoma. The accuracy of FNAC can reach more than 95% in adequate specimens [23].

The cytological features of TCV-PTC have been well described [24]. Nevertheless, preoperative diagnosis of TCV-PTC on FNAC is difficult, and it is more common for patients to be diagnosed postoperatively after histopathological examination [25]. Use of molecular testing and immunochemistry may aid in the preoperative diagnosis of TCV-PTC and other more aggressive variants of PTC: the BRAF V600E mutation is highly prevalent in TCV-PTC, with reports ranging from 66% to 100% [26,27,28,29,30,31]. The conventional smear is the standard diagnostic method for detecting thyroid lesions. Liquid-based cytology can improve detection of tall cells, because the cytoplasm of the cells is not well preserved in the conventional smear. Liquid-based cytology allows distinguishing TCV-PTC from PTC with tall cell features, and the residual material in fixative solution allows further studies to be carried out, such as immunostaining or molecular testing [32]. In 1976, Hawk and Hazard retrospectively reviewed 197 cases of PTC from 1921 to 1960. Four patients with a diagnosis of TCV-PTC died of the disease, and the mean follow-up was 7.3 years. The authors were the first to describe an association of TC cytology with larger tumors and older age [3]. In 2004, Sywak et al. reviewed 209 cases of TCV-PTC. The authors of this study found high rates of extracapsular spread of tumors (67%) and cervical adenopathy (57%); 25% of the patients showed locoregional recurrence and 22% developed distant metastases [33]. In the same year, Machens et al. found the association between TC cytology and distant metastases (50% in the TCV-PTC cohort vs. 31% in PTC) [34]. In 2007, Ghossein et al. reported that TCV-PTC without extrathyroidal extension has a more aggressive behavior than conventional intrathyroidal PTC, with a significantly higher nodal metastatic rate independent of tumor diameter, sex, and age [16]. In 2010, Jalisi et al. published a systematic literature review to evaluate the prognosis of TCV compared to cPTC. The TCV patients showed a higher rate of extrathyroidal extension (cumulative average of 60.3%), a higher rate of distant metastases at diagnosis (cumulative average of 15%), and nodal metastases (cumulative average of 58.1%). The cumulative average recurrence and the cumulative average disease-related mortality were higher in the TCV group versus cPTC group (42.5% vs. 9.8% and 23.6% vs. 1.5%, respectively) [35]. In a large multicenter study, Shi et al. confirmed the association between PTC-TCV and high-risk parameters, including extrathyroidal invasion, lymph node metastasis, stage III/IV, disease recurrence, mortality, and the use of radioiodine treatment [36].

In our study, the histopathological and prognostic data of patients with a diagnosis of TCV-PTC agreed with the data reported in previous studies. TCV-PTC showed a greater diameter than cPTC (2.69 vs. 2.22 cm), and angiolymphatic invasion, thyroid parenchymal invasion, extrathyroidal extension, and lymph node metastases were greater in TCV-PTC than cPTC (17.14% vs. 5.06%, 37.14% vs. 17.72%, 31.42% vs. 5.06%, and 45.71% vs. 13.29%, respectively). TCV-PTC also showed higher rates of local recurrence (17.1% vs. 6.3%) and distant metastases (5.71% vs. 0%). Five-year disease-free survival in patients with a diagnosis of PTC-TCV was 82.3% in the TCV-PTC group, whereas it was 92.3% in the cPTC group. All these findings reached statistical significance. Local recurrences in TCV-PTC patients were identified early in the follow-up. They all occurred within 16 months of surgery, while the few distant metastases occurred within 24 months. Similarly, local recurrences in the cPTC group were early, all occurring within 25 months of surgery. Clinical aggressiveness of TCV-PTC seems to be related to certain factors elaborated by the tumor. The high expression of Muc1 and type IV collagenase in TCV-PTC may allow for degradation of stroma, and this can be responsible for the greater invasiveness compared to usual and follicular variants of PTC [37,38]. The clinical behavior of TCV-PTC may also be related to the higher prevalence of activating point mutations of the BRAF compared to cPTC [38]. Indeed, tumors characterized by BRAF mutations in their molecular profile have a higher rate of extrathyroidal extension and nodal metastases and show a higher stage than BRAF-negative tumors [28]. The impact of TC cytology on prognosis requires a more aggressive therapeutic approach than typically followed in other PTC variants. However, the main problem is that TCV-PTC is diagnosed on histopathological examination after the initial thyroid surgery has been performed. If a patient is diagnosed with TCV-PTC and has undergone partial thyroidectomy, the patient should return to the operating room to perform a completion thyroidectomy associated with central neck dissection followed by RAI. If a total thyroidectomy is performed, the therapeutic choices to consider are the execution of central neck dissection followed by RAI versus RAI alone, and if the tumor is not iodocaptant, external beam radiation can be a therapeutic option [35]. Sywak et al. described that the treatment of choice of TCV-PTC is represented by total thyroidectomy associated with cervical lymph node dissection in the case of lymph node involvement, and en bloc resection of the perithyroidal tissues is necessary if their involvement is detected before or during surgery [33]. Prendiville et al. found that all patients with TCV-PTC require aggressive surgical treatment in association with RAI, levothyroxine suppressive therapy, and close follow-up [15].

In agreement with these findings, all patients enrolled in this study underwent total thyroidectomy, while more conservative surgery was not performed. Central compartment lymph node dissection and modified lateral neck dissection were performed more frequently in TCV-PTC patients than in cPTC patients (42.85% vs. 18.03% *p =* 0.001 and 17.14% vs. 6.96% *p =* 0.04, respectively). Consequently, the mean operative time increased in TCV-PTC patients (121 ± 29.01 min vs. 100 ± 28.91 min *p =* 0.0004), with no differences in postoperative stay between the two groups. In this study, the problem of difficult preoperative diagnosis of TCV-PTC emerged. In the case of postoperative diagnosis of TCV-PTC, no patient underwent rescue central compartment lymph node dissection because, in our clinical practice, total thyroidectomy was always associated with a central or modified lateral neck dissection in the case of clinical or ultrasound findings suggestive of lymph node involvement [39,40]. Total thyroidectomy is a safe procedure and showed a similar low complication rate in the two study groups. TCV-PTC patients should always be considered at intermediate risk of recurrence. Among the factors known to confer an intermediate risk of recurrence are extrathyroidal extension, lymph node metastases at the time of surgical treatment or I^131^ uptake outside the thyroid lodge on a post-therapy whole-body scan (Rx-WBS) performed after initial radioablative therapy [41,42], aggressive tumor histology, and vascular invasion [15,43,44]. All patients enrolled in this study underwent at least one course of RAI because it is always indicated when individual histology confers an intermediate risk of recurrence [45].

Our study has a main limitation. It is based on a retrospective analysis from a single institution. However, as ours is a high-volume center in a region where thyroid disease is endemic, our results are in line with those reported in the literature.

## 5. Conclusions

TCV is the most frequent aggressive histopathological variant of PTC. Our patients with a diagnosis of TCV-PTC showed larger tumor diameter and higher frequency of angiolymphatic invasion, extrathyroidal invasion, and lymph node metastases at the time of surgery than cPTC patients, and this resulted in a higher rate of local recurrence and distant metastases and in a less favorable clinical outcome. In this study, TC histology was therefore confirmed as an independent negative prognostic factor. Among the authors, there is the general consensus that the surgical treatment of TCV-PTC should be aggressive, with the execution of total thyroidectomy associated with central neck dissection followed by RAI. This good practice of therapeutic conduct is contrasted by TCV-PTC being largely underdiagnosed, being usually identified on definitive histopathological examination in patients with a generic preoperative diagnosis of suspicious or indeterminate nodule or PTC. We think that a good therapeutic option is to combine total thyroidectomy with the execution of central and/or modified lateral compartment lymph node dissection in the case of clinical or ultrasound findings suggestive of lymph node involvement, associated with RAI. Surgery is safe, and the complication rate is similar to that of cPTC patients. Further studies are needed to increase the chances of preoperative diagnosis of aggressive variants of PTC and to perform a tailored surgical treatment based on precise preoperative findings.

## Figures and Tables

**Figure 1 jcm-10-00070-f001:**
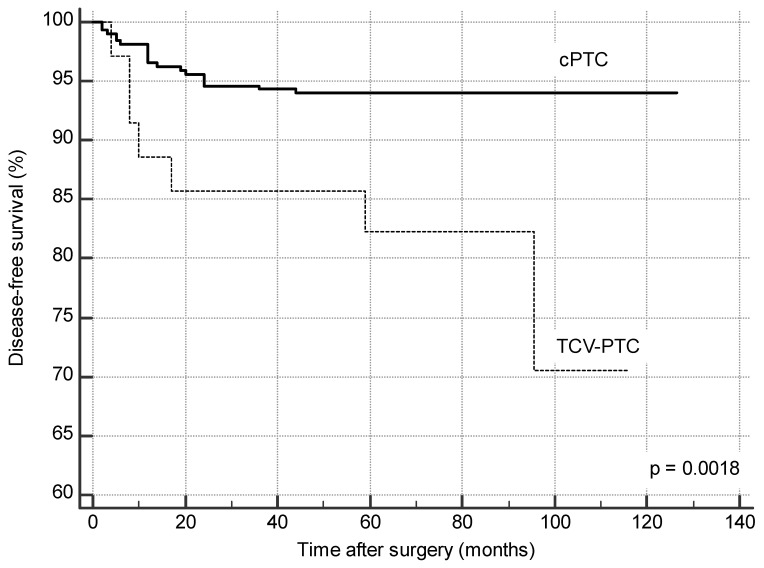
Disease-free survival (Kaplan–Meier method).

**Table 1 jcm-10-00070-t001:** Demographic and preoperative data.

	Group A (*n* = 35)	Group B (*n* = 316)	*p*-Value
Sex Male Female	14 (40%) 21 (60%)	80 (25.31%) 236 (74.68%)	0.07
Age (years, mean ± SD)	49.34 ± 18.22	50.62 ± 14.9	0.6
Indeterminate or suspicious nodule on cytology	8 (22.85%)	131 (41.45%)	0.04
Diagnosis of carcinoma on cytology	14 (40%)	24 (7.59%)	<0.0001
Suspicious nodule on US	14 (40%)	119 (37.65%)	0.8
Benign disease (multinodular goiter and hyperthyroidism)	10 (28.57%)	87 (27.53%)	0.8

**Table 2 jcm-10-00070-t002:** Surgical procedure and postoperative stay.

	Group A (*n* = 35)	Group B (*n* = 316)	*p*-Value
Total thyroidectomy and central compartment dissection	15 (42.85%)	57 (18.03%)	<0.001
Total thyroidectomy and modified lateral neck dissection	6 (17.14%)	22 (6.96%)	0.04
Surgical time (min, mean ± SD)	121 ± 29.01	100 ± 28.91	0.0004
Postoperative stay (days, mean ± SD)	2.74 ± 0.97	2.68 ± 0.93	0.7

**Table 3 jcm-10-00070-t003:** Histopathological diagnosis.

	Group A (*n* = 35)	Group B (*n* = 316)	*p*-Value
Tumor size (cm, mean ± SD)	2.69 ± 1.27	2.22 ± 1.25	0.03
Thyroid weight (grams, mean ± SD)	31.9 ± 42.99	33.91 ± 42.37	0.7
Multicentric carcinoma	11 (31.42%)	100 (31.64%)	1
Angiolymphatic invasion	6 (17.14%)	16 (5.06%)	0.01
Carcinoma infiltrating the glandular parenchyma	13 (37.14%)	56 (17.72%)	0.004
Extrathyroidal extension	11 (31.42%)	16 (5.06%)	<0.0001
Node metastases	16 (45.71%)	42 (13.29%)	<0.0001

**Table 4 jcm-10-00070-t004:** Postoperative complications.

	Group A (*n* = 35)	Group B (*n* = 316)	*p* Value
Postoperative hematoma	1 (2.85%)	4 (1.26%)	0.4
Wound infection	0	0	
Transient recurrent laryngeal nerve palsy	1 (2.85%)	4 (1.26%)	0.4
Permanent recurrent laryngeal nerve palsy	0	0	
Transient hypoparathyroidism	11 (31.42%)	87 (27.53%)	0.6
Permanent hypoparathyroidism	3 (8.5%)	20 (6.3%)	0.6
Chylous fistula	0	0	

**Table 5 jcm-10-00070-t005:** Local recurrence and distant metastases.

	Group A (*n* = 35)	Group B (*n* = 316)	*p*-Value
Local recurrence	6 (17.1%)	20 (6.3%)	0.02
Distant metastases	2 (5.71%)	0	0.009
Follow-up (months, mean ± SD)	79.4 ± 25.9	98.4 ± 26.5	0.09

## Data Availability

Data will be made available under reasonable request.

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
