# Peer review of "Tall Cell Variant versus Conventional Papillary Thyroid Carcinoma: A Retrospective Analysis in 351 Consecutive Patients"

_jcm, 2020, doi:10.3390/jcm10010070_

Round 1

Reviewer 1 Report

Dear Authors, 

I've read with interest the manuscript proposed for publication, reporting good retrospective evaluation of differentiated  thyroid carcinomas, focusing on tall cell variant features. 

I have nothing to add, I only suggest a text revision with specific check for english grammar.

Author Response

Dear Reviewer 1,

thank you for your analysis of our manuscript and your constructive suggestion.

We have improved the English grammar as you suggested.

Best regards

Reviewer 2 Report

Alessandro et al. provide a clinical comparison between patients with conventional PTC and patients with tall cell variant of PTC at their institution. They conclude that the tall cell variant has a more aggressive course. This has already been established in the literature for quite some time and thus the study lacks novelty. There may be some utility in publishing institution-specific data on this topic, however, and the report is well-written and provides an excellent review of the tall cell variant of PTC, which may be useful to the journal readership.

Some comments:

  1. The report generally lacks novelty. Could the authors clarify whether there was any novel finding in their study and, if so, emphasize this?
  2. Some sections are not broken into paragraphs which makes the manuscript difficult to read. Please break these sections into smaller paragraphs.
  3. For the text on TCV detected by FNA, the authors have cited reference 20. They may wish to read and/or cite https://doi.org/10.1159/000353823 which is the most relevant reference regarding TCV in FNA specimens.
  4. Overall, the manuscript is concisely and clearly written and I enjoyed reading it.

Author Response

Dear Reviewer 2,

thank you for your analysis of our manuscript and your constructive suggestion.

We have improved the English grammar as you suggested.

  1. Indeed in our study, the data reported are in line with those previously published in the literature. We think that a strong element is the clear selection of patients based on the histological subtype, an element lacking in many similar studies. Also, our study reports the results of a single-center, but with high volume, in a region where thyroid disease is endemic.
  2. As you suggested, we have divided the "materials and methods" section into paragraphs.
  3. We read the article you suggested with great interest and included it among the references.

Corrections in the text are highlighted in green.

Best regards 

Reviewer 3 Report

The author aimed to analyze the clinical and pathologic characteristics of tall cell variant papillary thyroid cancers.

Major comments

There are already substantial numbers of similar studies including meta-analysis. It is hard to find the differences in results of this study from already known characteristics in prior studies. Could authors suggest different findings from other studies? Could authors present strength and limitation of your study compared to others?

In addition, study design is unfocused. In the introduction, there was no comment on surgical procedures and complications. But, in the study, surgical procedure and complications presented as one of main outcomes. What is the main purpose of the study?

How many pathologists were involved in the diagnosis of TV PTC? Did authors review the pathology of all cases to confirm subtypes of PTCs?

Did authors have information about BRAF mutation status?

Table 2 should be modified. Especially, it is hard to understand the meaning of "Diagnosis of carcinoma on cytology" in table 2. According to the results, most of patients enrolled in the study underwent surgery with evidence of malignant cytologic results. Then, what were the reasons for the surgery?

“Method” should be shortened and “Discussion” should be focusing on findings of study.

Minor comments

There are many errors in writing. (ex. expression of p value, percentage and spells)

Author Response

Dear Reviewer 3,

thank you for your analysis of our manuscript and your constructive suggestion.

Answers to major comments:

Indeed in our study, the data reported are in line with those previously published in the literature. We think that a strong element is the clear selection of patients based on the histological subtype, an element lacking in many similar studies. In addition, our study reports the results of a single-center, but with high volume, in a region where thyroid disease is endemic. We have added comments related to the limitations of our study at the end of the "discussion" section.

The main purpose of our study is the analysis of the clinical and histopathological characteristics of TCV-PTC. The analysis of the surgical procedure and complications is not the main outcome of the study.

In our institute, the histopathological diagnosis of thyroid diseases is assigned to a small working group, specialized in this field. For this reason, we did not need to review the pathology.

Unfortunately, we have no information on the status of the BRAF mutation. As the study is retrospective and based on an institutional database, we are unfortunately unable to retrieve this information.

With "diagnosis of carcinoma on cytology" we indicated the result "Tir5" on cytological examination. The other surgical indications arise from the detection of suspicious or indeterminate nodules on cytology (Tir4 or Tir3). Further indications arise from the presence of nodular goiter or a condition of hyperthyroidism. The results of the cytological analysis are stratified according to the 2014 SIAPEC classification.

To make the "materials and methods" section clearer, we have indicated subsections.

Answers to minor comments:

We have improved the English grammar as you suggested and corrected the writing errors.

Corrections in the text are highlighted in green.

Best regards

Reviewer 4 Report

In  this retrospective study the authors investigate the clinical and pathological features of two papillary thyroid carcinoma (PTC) variants; they assess the conventional PTC variant, the more frequent among PTCs, compared to tall cell variant, a quite rare and aggressive PTC variant.

This is a descriptive study and the presented data confirm the aggressive nature of TCV PTC already reported by several previous publications. The study could be of some interest considering an epidemiological point of view, eventhough it represents a Short Communication more than an Original article as a limited amount of data and analyses are reported.

Some improvements are required.

- The classification in group A and B is not useful and confounding, also for the authors who assign the wrong group label in lines 92 (group A) and in line 96 (group B).

The recognized acronyms conventional PTC (cPTC) and tall cell variant PTC (TCV-PTC), also used by the same authors, are already meaningful definitions and should be used as group labels instead of the generic and confounding A and B across all manuscript and tables.

- Data are unnecessarily divided in 5 separate tables. A unique table can be provided using subheadings, for an example see DOI: 10.1089/thy.2013.0503

-References are not always up to date, more recent and international studies on big patients series are available and should be included as discussed, for instance

DOI: 10.1089/thy.2013.0503, Thyroid 2014

DOI: 10.1210/jc.2015-2917, JCEM 2016

 - In the Discussion (line 138-140) the authors state: “ tumors with less than 50% TC are excluded by diagnosis of TCV-PTC and were excluded from this study because the implications of the presence of small numbers of tall cells in a thyroid tumor are actually unknown [6,7,8,9]”.

This is not correct. More recent publications have showed that the presence of either focal TC, or TC features, or small TC component (≥10%) confers aggressive features.

DOI: 10.1530/EC-18-0333, Endocr Connect. 2018

DOI: 10.1245/s10434-019-07444-2, Ann Surg Oncol. 2019

DOI: 10.1016/j.surg.2013.05.009, Surgery 2013

Could be of interest and a more appropriate and updated analysis to include in the manuscript the assessment of the excluded PTC with TC features or small percentage of TC component.

- Results as presented are a mere repetition of tabular data

- Age (years), Surgical time (min), Postoperative stay (days), Tumor size (cm), Thyroid weight (grams) ,

which is the reported value? Mean, median? and the ± error? is it Standard Deviation? This should be clearly indicated

- AJCC staging (Stage I, II,III, or IV) is missing

- line 74: various ATA guidelines exist according to different years, the reference of the considered ATA guidelines should be included.

-Typos lines 38-39

Author Response

Dear Reviewer 4,

thank you for your careful analysis of our manuscript and your constructive suggestion.

We have corrected the errors reported. However, we believe that our nomenclature (group A and B, described in the “materials and methods” section) is overall clear and easy to interpret and that it avoids the repetition of more complex acronyms.

To make the interpretation of our data easier and more practical, we preferred to use separate tables inserted in the text. A single table containing all the data could be difficult to consult.

We have read with interest the articles you suggested and we have included them in our references. However, we specify that the analysis of the characteristics of the PTC with tall cell features is not a primary endpoint of our work.

As you suggested, we have indicated the results more clearly (see corrections in the tables).

Our endocrinologists use the most up-to-date ATA guidelines available. In this case, the ATA guidelines used were those of the year 2009. We have corrected the references.

We did not include the AJCC staging system because, as known, this kind of staging is related to overall survival but not to disease-free survival. For this reason, considering that thyroid cancer is a tumor with an overall survival comparable to that of the general population, including the AJCC staging system in our paper would be redundant and would not add significant issues to the work.

We have corrected typos in lines 38-39.

Corrections in the text are highlighted in green.

Best regards

Round 2

Reviewer 2 Report

The authors have appropriately responded to the comments.

Reviewer 4 Report

The work has been improved according to reviewers’ suggestions; some corrections have been made.